# New Synthetic Opioids: What Do We Know About the Mutagenicity of Brorphine and Its Analogues?

**DOI:** 10.3390/ijms26115084

**Published:** 2025-05-26

**Authors:** Monia Lenzi, Sofia Gasperini, Sabrine Bilel, Giorgia Corli, Francesca Rombolà, Patrizia Hrelia, Matteo Marti

**Affiliations:** 1Department of Pharmacy and Biotechnology, Alma Mater Studiorum University of Bologna, 40126 Bologna, Italy; sofia.gasperini4@unibo.it (S.G.); francesca.rombola3@unibo.it (F.R.); patrizia.hrelia@unibo.it (P.H.); 2Department of Translational Medicine, Section of Legal Medicine, LTTA Center and University Center of Gender Medicine, University of Ferrara, 44121 Ferrara, Italy; sabrine.bilel@unife.it (S.B.); giorgia.corli@unife.it (G.C.); matteo.marti@unife.it (M.M.); 3Clinical Pharmacology Unit, IRCCS AOU Bologna, 40138 Bologna, Italy; 4Collaborative Center for the Italian National Early Warning System (NEWS-D), Department of Anti-Drug Policies, Presidency of the Council of Ministers, 00186 Rome, Italy

**Keywords:** Brorphine, novel synthetic opioids, new psychoactive substances, flow cytometry, genotoxicity, in vitro mammalian cell micronucleus test, TK6 cells

## Abstract

Since 2019, a growing number of structurally diverse, non-Fentanyl-related novel synthetic opioids (NSOs) have emerged, but little is still known on the toxic profile of several of the molecules belonging to this class. Regarding long-term toxicity, few studies have investigated the genotoxic potential of NSOs, and no genotoxic data at all are available for the subclass of Brorphine-like benzimidazolone opioids. To deepen and broaden our understanding of their toxicological profile, this study was aimed at evaluating the genotoxicity of Brorphine and four of its analogues (Orphine, Fluorphine, Chlorphine and Iodorphine) on human lymphoblastoid TK6 cells employing a flow cytometric protocol of the “In Vitro Mammalian Cell Micronucleus (MN) test”. The results show a statistically significant MNi increase for Fluorphine, Chlorphine and Iodorphine, but not for Brorphine and Orphine, demonstrating for the first three the ability to induce chromosomal damage. Afterwards, Brorphine and Orphine were tested on TK6 cells also in the presence of an exogenous metabolic activation system (S9 mix) to consider the possible genotoxic hazard posed by their metabolites as well. Also, under this experimental condition, no statistically significant increase in the MNi frequency was detected.

## 1. Introduction

New psychoactive substances (NPS) are continually emerging in the drug market. Until recently, novel synthetic opioids (NSOs), considered some of the most hazardous among these substances, mainly consisted of analogues of the powerful painkiller Fentanyl. Since 2019, a growing number of structurally diverse, non-Fentanyl-related NSOs have emerged [1]. In August 2019, the DEA reported the first appearance of Brorphine (often marketed on the darknet as “purple heroin”) in the recreational opioid market in the United States, just one year after its synthesis by N. Kennedy, the scientist behind a series of functionally selective opioid compounds. Initially synthetized and patented in the 1960s by Janssen Pharmaceuticals, Brorphine is an N-benzylpiperidinyl derivative of benzimidazol-2-one (1-{1-[1-(4-bromophenyl) ethyl] piperidin-4-yl}-1,3-dihydro-2H-benzimidazol-2-one), and belongs to a class of MOR agonists that have a significant signaling bias in G protein signaling [2]. In assays of agonist activity (G protein activation) at the MOR, Brorphine has proven to be a high-efficacy agonist, approximately 13 times more potent than morphine [3].

Following its initial identification, the presence of Brorphine escalated rapidly. Between June and July 2020, 20 deaths involving confirmed Brorphine exposure were reported in the United States [1]. In early 2020, Brorphine officially entered the European Union’s drug market, with its presence reported in four countries: Belgium, Germany, Slovenia, and Sweden [2]. In Slovenia, a user submitted two counterfeit Oxycodone tablets containing Brorphine for analysis. The user had purchased the tablets from the darknet drug market and experienced severe adverse effects. Although the poisoning was deemed non-life-threatening, hospital treatment was necessary. Clinical symptoms of the poisoning included prolonged loss of consciousness, rhabdomyolysis, and acute kidney failure [4].

Due to its dangerous and alarming effects, the DEA temporarily classified Brorphine as a Schedule I substance under the Controlled Substances Act in December 2020, with the decision becoming permanent in March 2021 [5]. Following the announcement of this scheduling action, instances of Brorphine use began to decline. Its life cycle on the market appeared to last approximately 12 months, with peak consumption spanning about six months. This is comparable to, or slightly shorter than, the cycle of Isotonitazene, highlighting the rapid turnover of NPS on the market [6].

To date, Brorphine remains the only NSO featuring a benzimidazolone scaffold. Although the NSO market is currently dominated by benzimidazole-based “nitazene” opioids [7], increasing awareness of the risks associated with nitazenes, along with the broader implementation of legislative measures [2], could drive a transition toward newer generations of NSOs. The emergence of new generations of NSOs, including Brorphine analogues, can be anticipated. A recent elegant study has focused on the synthesis and pharmacological characterization of Brorphine and four structurally similar benzimidazolone opioids (Orphine, Fluorphine, Chlorphine, and Iodorphine). In this work, in vitro evaluations (including MOR binding affinity and activation) and in vivo assessments (such as antinociception and respiratory depression) were performed to investigate the opioid-like properties of these compounds. This study revealed that the analogues with smaller substituents (Fluorphine, Orphine) demonstrated the highest MOR affinity, while Chlorphine, Brorphine, and Iodorphine were generally the most active in terms of in vitro MOR activation, as measured by βarr2 and mini-Gαi recruitment potential [2]. In vivo, the antinociceptive and respiratory effects of these drugs were assessed in male CD-1 mice. All analogues increased the threshold for acute mechanical and thermal pain stimuli, but only Brorphine and Chlorphine achieved maximum antinociception in both pain assays [2]. Most of the analogues also had negative effects on respiration, including reductions in respiratory rate, breath length, and tidal volume. The findings contribute to understanding the potential for Brorphine-like benzimidazolone opioids to emerge as NSOs [2]. NSOs, in particular Fentanyl analogues, were found to be genotoxic [8]. Studies on the genotoxicity of Brorphine and its analogues are not yet available. This study is aimed at evaluating the genotoxicity of Brorphine and its four analogues (Orphine, Fluorphine, Chlorphine, and Iodorphine) (Figure 1) in human lymphoblastoid TK6 cells, particularly in terms of their ability to induce structural and numerical chromosomal aberrations. For this purpose, we evaluated the frequency of micronuclei (MNi) by employing a flow cytometric (FCM) protocol developed in our laboratory and published previously [8].

## 2. Results and Discussion

According to OECD guideline No. 487, to evaluate the mutagenicity of a xenobiotic, the highest concentration tested should induce a cytotoxicity not exceeding 55 ± 5%, and consequently must allow a cell viability of at least 45 ± 5% [9].

Therefore, in the first phase of the research, we checked the cytotoxicity induced by the molecules under study in order to establish the concentrations to be used in the subsequent assays. In particular, the cytotoxicity induced by Orphine, Brorphine, Chlorphine, Iodorphine and Fluorphine was evaluated after 26 h of treatment (time necessary for TK6 cells to complete approximately 1.5–2 replication cycles), measuring the viability at the different concentrations tested (0–100 µM) and then normalizing to that obtained in the untreated control cultures (0 µM).

Figure 2 shows that viability remains well above the threshold required by the OECD (represented by the red line) at all Orphine and Fluorphine concentrations tested—up to 50 μM for Brorphine and Iodorphine and up to 75 μM for Chlorphine.

In addition, to ensure the reliability of a mutagenesis analysis, it is crucial to assess cell proliferation to confirm that an adequate number of cells have undergone mitosis. This ensures that the mutagenesis test is conducted under appropriate levels of cytotoxicity also in terms of cytostasis. In fact, according to OECD guideline No. 487, “it is necessary to demonstrate that the cells in culture have divided, ensuring that a substantial proportion of the cells analyzed have undergone cell division during or after a treatment period equivalent to 1.5–2 replication cycles.” Like the approach for cytotoxicity, the guideline establishes a threshold of 55 ± 5% for cytostasis and recommends calculating the Relative Proliferation Doubling (RPD) as an indicator of cell proliferation. Accordingly, the RPD must be at least 45 ± 5% [9].

The results obtained demonstrate a different behavior of the molecules under study. Indeed, for Orphine, all concentrations tested remained above this threshold, while for Brorphine, Iodorphine and Fluorphine, cytostasis was acceptable up to 25 µM. For Chlorphine, the highest acceptable concentration was 12.5 µM (Figure 3).

To note, cell cycle arrest could represent an important genotoxicity-related parameter. In fact, healthy cells facing DNA damage should slow down their replication speed while activating the DNA damage response machinery in an effort to either repair DNA before cell duplication or induce apoptosis in case of irreparable damage [10].

Accordingly, OECD guideline No. 487 also suggests evaluating other cytotoxicity markers including cell integrity, necrosis and apoptosis to ensure a comprehensive and accurate determination of the concentrations to be tested for mutagenesis evaluation [9]. Therefore, the research proceeded by investigating the potential induction of apoptosis by the substances under study, at concentrations selected based on cell viability and RPD.

The evaluation of the apoptotic process through double-staining with Annexin V Alexa Fluor 488 and PI shows that, in the case of Orphine, the induction of apoptosis remained similar to that measured in the negative control at concentrations of 12.5, 25, and 50 µM, while it doubled compared to the negative control at concentrations of 75 and 100 µM (Figure 4A). Therefore, these last two concentrations were excluded from the MN test. As for Brorphine, Chlorphine, Iodorphine, and Fluorphine, the increase in the percentage of apoptotic cells never exceeded double the levels observed in the control cultures (Figure 4B–E).

It is acknowledged that genetic damage can trigger different cellular responses, among which we can include apoptosis and related processes. In this case, the impaired cells are eliminated before they can replicate and create a clone of the mutated cells [10]. In this regard, it is worth noticing how the molecules under study, except for Orphine, are all unable to induce apoptosis at the concentrations tested, which can represent quite an alarming characteristic if accompanied by a positive genotoxic outcome.

A possible explanation for why Orphine activated apoptosis unlike other Brorphine derivatives might lie in the higher concentrations tested, selected based on the previous PI assay. Maybe such concentrations are not able to completely block proliferation, and thus induce necrosis or advanced stages of apoptosis, after 26 h of treatment, but can induce early stages of apoptosis detectable by the Annexin V assay.

Based on the results obtained from the cytotoxicity, cytostasis, and apoptosis assays, the concentrations for the mutagenesis test were selected. Specifically, Orphine was tested at 12.5, 25 and 50 µM; Brorphine, Iodorphine, and Fluorphine at 6.25, 12.5 and 25 µM; and Chlorphine at 3.12, 6.25 and 12.5 µM.

To evaluate the potential mutagenic effect, the MNi frequency was measured in untreated cultures (negative controls), in those treated with the different Brorphines derivatives under study, and in cultures treated with MMC and VINB, mutagens used as positive controls. The MN test was conducted by FCM using the automated method published by Lenzi et al. [11]. This protocol offers numerous advantages over the classic method of optical microscopy, since, by analyzing a number of events five times higher, it allows robust statistical analysis and enables one to identify even the so-called weak genotoxics. Additionally, it guarantees a much more objective result not affected by the subjectivity of interpretation of the operator, as inevitably happens in optical microscopy, and, among other things, with a notable reduction in analysis times [11].

This method allowed us to demonstrate the mutagenic capacities of Chlorphine, Iodorphine and Fluorphine, which, in fact, have shown a statistically significant increase in the frequency of MNi at 12.5, 25 and 12.5 concentrations, respectively. On the contrary, Orphine and Brorphine were not found to be mutagenic in terms of the ability to induce structural or numerical chromosomal aberrations (Figure 5). These diverse outcomes underline how assessing the genotoxic hazard only for a few representative compounds of a structural class of molecules is often not sufficient, and that every single analogue should be tested. In fact, as in some of our previous studies on NPS, different biological effects were observed within the same structural class [12,13]. In particular, in one study, we addressed the assessment of the potential genotoxicity of Fentanyl and three Fentanyl-related opioids, demonstrating the non-genotoxicity of Fentanyl, i.e., the pharmaceutical progenitor of the class, while its illicit non-pharmaceutical analogues (Acrylfentanyl, Furanylfentanyl and Ocfentanyl) were found to be genotoxic [8].

Moreover, it is fundamental to consider the potential mutagenicity of any metabolites of the substances under study as well, and, in this regard, it is crucial to underline that TK6 cells have a rather limited metabolic capacity per se. For this reason, the guidelines for mutagenicity assessment suggest that a negative result should be followed by a test carried out in the presence of an extrinsic source of metabolic activation for short-term treatment. In this study, a mixture containing hepatic microsomal enzymes added with the necessary cofactors (S9 mix) was employed to this end, i.e., to completely exclude or prove the mutagenic capacity associated with Orphine and Brorphine [9]. The treatment time selected for TK6 cells was 3 h, followed by 23 h of recovery in fresh medium, as recommended by the OECD guideline [9]. Orphine and Brorphine were tested at the same concentrations as were selected for the 26 h long-term treatment (Orphine 12.5, 25 and 50 µM; Brorphine 6.25, 12.5 and 25 µM). Nevertheless, albeit after a much shorter treatment time, viability, RPD and apoptosis thresholds were also checked under this experimental condition.

Figure 6 displays how Orphine and Brorphine in the presence of the S9 mix still give a negative outcome in the MN test.

Future investigations could be aimed at further assessments of the genotoxic potential of Orphine and Brorphine. In fact, for a comprehensive evaluation of the genotoxic hazard and to confirm a negative outcome, an additional assay for genic mutations (e.g., a bacterial reverse mutation test) should be performed, as the MN test only evaluates the ability to cause structural and numerical chromosomal aberrations. On the contrary, in the case of an already positive response for other validated genotoxicity endpoints, e.g., the in vitro mammalian cell MN test, supplementary assays are not required, as for Chlorphine, Iodorphine and Fluorphine [9,14].

The absence of any in vivo experiments in the present study could seem to limit its translational potential, and therefore represent its major limitation. Certainly, data from in vivo studies would strengthen the results of this study, but such data are currently not available in the literature. However, it is important to point out that the assessment of a substance’s mutagenicity is one of the few toxicological endpoints for which several in vitro tests have been validated. Furthermore, many regulations agree that in the case of positive results obtained in vitro, these may be considered sufficient, in line with the 3Rs principle and the aim of reducing animal testing as much as possible. In contrast, in vivo experiments are clearly necessary to confirm negative in vitro results, following a precautionary approach to protect human health [14].

Moreover, it could be asked whether the concentrations we tested are reachable in vivo, and if they are comparable to the level of Brorphines detectable in human users, so as to clarify how these concentrations relate to realistic human exposure scenarios and support the biological relevance of our findings. Unfortunately, information regarding levels of human exposure to Brorphines is currently not available. However, it is recognized that it is not possible to define a No-Observed-Adverse-Effect Level (NOAEL) for genotoxic substances, and zero risk corresponds only to the zero dose; therefore, any dose is potentially toxic. When increasing the dose and the exposure, the likelihood that damage will occur increases [12].

Lastly, to check whether the evident inhibition of cell proliferation (reduction in RPD value) observed after a 26 h treatment was followed by actual cell death and therefore a stronger and more time-dependent cytotoxic effect, the cell viability and proliferation induced by these Brorphine derivatives could also be evaluated after a longer treatment time.

## 3. Materials and Methods

### 3.1. Reagents

7-aminoactinomycin (7-AAD), annexin V-phycoerythrin (Annexin V-PE), cyclophosphamide (CP), dimethyl sulfoxide (DMSO), ethylenediaminetetraacetic acid (EDTA), fetal bovine serum (FBS), L-glutamine (L-GLU), mitomycin C (MMC), Nonidet, penicillin–streptomycin solution (PS), phosphate-buffered saline (PBS), potassium chloride, potassium dihydrogen phosphate, propidium iodide (PI), Roswell Park Memorial Institute (RPMI) 1640 medium, BPC-grade water, sodium chloride, sodium hydrogen phosphate, vinblastine (VINB) (all purchased from Merck, Darmstadt, Germany), RNase A, SYTOX Green (purchased from Thermo Fisher Scientific, Waltham, MA, USA) and Mutazyme 10% S9 mix (purchased from Moltox, Boone, NC, USA) were used.

### 3.2. Brorphine and Its Analogues

Orphine, Brorphine, Chlorphine, Iodorphine and Fluorphine were purchased from Cayman Chemical (Ann Arbor, MI, USA). The molecules were dissolved in absolute ethanol at a maximum concentration of 10 mM and stored at −20 °C. To avoid potential solvent toxicity, the concentration of absolute ethanol was kept below 1% *v*/*v* for all the experimental conditions.

### 3.3. Cell Culture

All experiments were carried out on human lymphoblastoid TK6 cells. OECD guideline No. 487 specifies that only rodent cell lines such as CHO, V79, CHL/IU, and L5178Y cells, or human cell lines such as TK6, are suitable for analyzing the presence of MNi to obtain reliable and robust results. Among the possible OECD-validated cells, the TK6 cell line was selected because of its human and non-tumoral origin, ease of maintenance in culture and replicative speed [9,15].

TK6 cells were purchased from ATCC (Manassas, VA, USA) and were grown at 37 °C and 5% CO_2_ in complete medium consisting of RPMI-1640 supplemented with 10% FBS, 1% L-GLU and 1% PS. Considering the doubling time of TK6 cells (13 h), the exponential growth of the cell culture was maintained by diluting cells every two days in fresh medium. Cell density never exceeded the critical value of 9 × 10^5^ cells/mL.

### 3.4. Test Conditions

TK6 cells underwent a continuous 26 h treatment with the test items. The total of 26 h corresponds to approximately 1.5–2.0 normal cell-cycle lengths for TK6 cells, as suggested by OECD guideline No. 487 [9].

Additionally, to evaluate the effects of both the parental compound and the related metabolites in vitro, TK6 cells were treated with selected Orphine and Brorphine in the presence of an exogenous metabolic activation system, as indicated by OECD guideline No. 487. Specifically, in these experiments, S9 mix was employed as the metabolizing system, a cofactor-supplemented post-mitochondrial fraction containing a liver enzymatic cocktail derived from rats treated with enzyme-inducing agents. The final concentration of S9 mix in the culture medium was 1%.

Since the OECD guideline recommends limited cell exposure to S9 mix, the experiments with S9 mix were performed after a short treatment time of 3 h + S9 mix, followed by 23 h of recovery in fresh medium [9].

#### 3.4.1. Selection of Concentrations

To identify the adequate concentrations to be tested for the MNi frequency evaluation, the OECD guideline No. 487 defined a threshold for cytotoxicity and cytostasis equal to 55 ± 5% compared to the negative control. Consequently, viability and proliferation should be at least equal to 45 ± 5% [9].

#### 3.4.2. Measurement of Cytotoxicity

To assess cytotoxicity, aliquots of 2.5 × 10^5^ TK6 cells were seeded and treated with Orphine, Brorphine, Chlorphine, Iodorphine or Fluorphine in the concentration range 0–100 μM for 26 h without S9 mix, or for 3 h with S9 mix followed by 23 h of recovery. At the end of the treatments, cytotoxicity was assessed by considering the viability percentage. More specifically, cells were stained with PI dye, which allows us to distinguish live cells from necrotic cells and emits in red. The viability was automatically calculated, acquiring 1000 events (cells) using the Guava software guavaSoft™ 4.5 (Cytek Biosciences, Inc., Fremont, CA, USA). For each sample, the viability percentage was recorded and normalized to the concurrent negative control, set at 100%.

#### 3.4.3. Measurement of Cytostasis

Aliquots of 2.5 × 10^5^ TK6 cells were treated with Orphine, Brorphine, Chlorphine, Iodorphine or Fluorphine in the concentrations range 0–100 μM for 26 h without S9 mix, or for 3 h with and without S9 mix followed by 23 h of recovery, and cytostasis was checked as follows. The numbers of seeded cells and those present at the end of the treatment, automatically calculated by the Guava software guavaSoft™ 4.5 (Cytek Biosciences, Inc., Fremont, CA, USA), were used to check cell replication by calculating population doubling (PD) (Equation (1)):(1)$$PD=\log\left(\frac{post-treatment\;cell\;number}{initial\;cell\;number}\right)÷log2$$

Thereafter, relative population doubling (RPD) was obtained by comparing the PD values calculated for negative controls to that of treated cultures to verify that the majority of cells had completed cell division after the treatment (Equation (2)).(2)$$RPD=\frac{PD\;in\;treated\;cultures}{PD\;in\;control\;cultures}\times 100$$

#### 3.4.4. Measurement of Apoptosis

To better select the concentrations to be tested in the MNi frequency analysis, the possible activation of the apoptotic process was evaluated as well, as one of the cell death mechanism alternatives to necrosis. In particular, the concentrations able to induce a two-fold increase in apoptosis compared to those registered in negative controls were excluded from the subsequent MNi frequency analysis.

Based on cytotoxicity and cytostasis results, aliquots of 2.5 × 10^5^ TK6 cells were treated with the selected concentrations in the range 0–100 μM for 26 h without S9 mix, or for 3 h with S9 mix followed by 23 h of recovery.

At the end of the treatments, cells were stained with the two fluorophores 7-AAD, which allows us to distinguish live cells from necrotic cells and emits in red, and Annexin-V-PE, which allows us to quantify the apoptotic cells through a yellow emission. For this assay, 2000 events (cells) were considered.

The Guava software guavaSoft™ 4.5 (Cytek Biosciences, Inc., Fremont, CA, USA) automatically calculates the percentage of live, apoptotic and dead cells. The apoptotic cell percentage recorded in treated samples was normalized to that registered in the negative control, set at 1, and expressed as apoptotic fold increase.

#### 3.4.5. Measurement of MNi Frequency

For the long-term treatment of 26 h without S9 mix, aliquots of 2.5 × 10^5^ TK6 cells were treated with different test item concentrations selected based on the results of cytotoxicity, cytostasis and apoptosis. In more detail, Orphine was tested at 12.5, 25 and 50 µM, Brorphine, Iodorphine, and Fluorphine at 6.25, 12.5 and 25 µM, and Chlorphine at 3.12, 6.25 and 12.5 µM concentrations.

For the treatment with S9 mix (3 h + S9 mix, followed by 23 h recovery), the selected Orphine and Brorphine that produced a negative result after the 26 h treatment without S9 mix were tested at the same concentrations for 26 h.

MMC and VINB, known clastogen and aneuploidogenic agents, respectively, were used as positive controls in the absence of S9 mix. On the contrary, the clastogen CP was used as a positive control that requires metabolic activation when in the presence of S9 mix [9].

At the end of the treatment, the MNi frequency was assessed by a flow cytometric protocol [11]. Briefly, aliquots of 7 × 10^5^ cells were collected and stained with two fluorophores 7-AAD, which allowed us to distinguish live cells from necrotic cells and emit in red and SITOX Green, allowing us to identify nuclei and micronuclei and emit in green. The MNi frequency was calculated as the number of MNi per 5000 nuclei derived from viable and proliferating cells [11]. The MNi frequency recorded in treated cultures was then normalized to that recorded in the concurrent negative control cultures, set equal to 1, and expressed as MNi frequency fold increase.

#### 3.4.6. Flow Cytometry

FCM analyses were carried out with a Guava EasyCyte 5HT Flow Cytometer II generation system equipped with a class IIIb laser operating at 488 nm (Cytek Biosciences, Inc., Fremont, CA, USA).

#### 3.4.7. Statistical Analysis

All assays were repeated in three independent experiments and the results are expressed as the mean ± Standard Error of the Mean (SEM). Statistical significance was analyzed by paired analysis of variance for repeated measures (repeated measures ANOVA), followed by Dunnett or Bonferroni post-test using Prism Software 9.0 (GraphPad Software, Boston, MA, USA). The results were considered statistically significant for *p* > 0.05. All statistical analyses were carried out according to the GraphPad Prism manual.

## 4. Conclusions

Overall, this study highlights a different cytotoxic capacity of the molecules under investigation, both in terms of induction of necrosis and apoptosis, and a marked cytostatic effect attributable to Brorphine, Clorphine, Iodorphine, and Fluorphine. Moreover, the most important finding is the proven mutagenic capacity of Chlorphine, Iodorphine and Fluorphine and the consequent possibility of serious long-term effects. On the contrary, neither Orphine and Brorphine, nor their corresponding metabolites, exhibited any mutagenic effect.

## Figures and Tables

**Figure 1 ijms-26-05084-f001:**
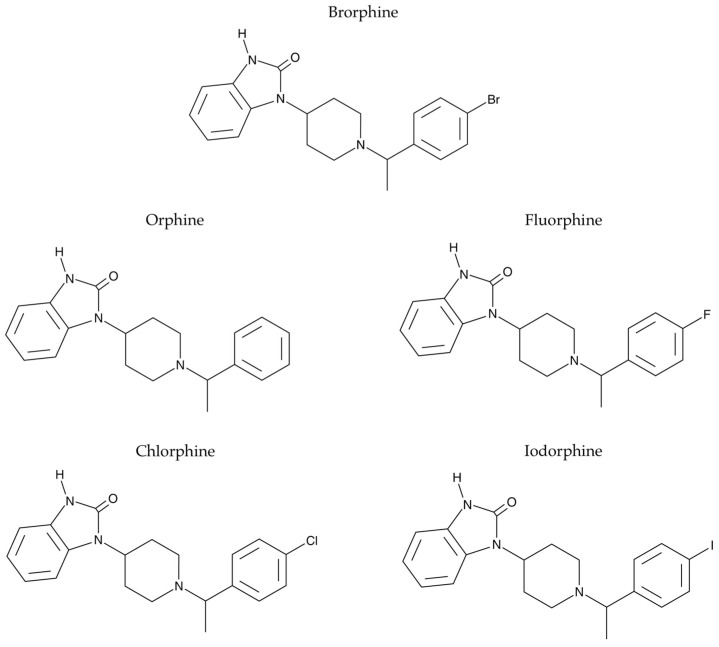
Chemical structure of Brorphine and its analogues Orphine, Fluorphine, Chlorphine, and Iodorphine.

**Figure 2 ijms-26-05084-f002:**
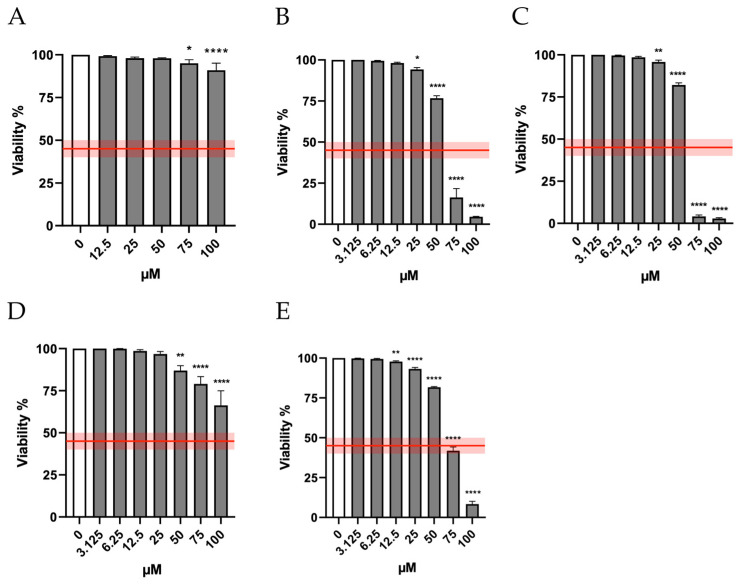
Viability of TK6 cells after 26 h treatment with (**A**) ORF, (**B**) Br-ORF, (**C**) I-ORF, (**D**) F-ORF and (**E**) Cl-ORF at the concentrations reported compared to the untreated negative control (0 µM). Each bar represents the mean ± SEM of at least three independent experiments. Data were analyzed by Repeated Measures ANOVA—(**A**) F (5, 19) = 8.494; *p* = 0.0002; (**B**) F (8, 19) = 685.7; *p* < 0.0001; (**C**) F (8, 18) = 2003; *p* < 0.0001; (**D**) F (8, 24) = 27.78; *p* < 0.0001; (**E**) F (9, 20) = 2318; *p* < 0.0001—followed by Dunnett post-test. The red line indicates the OECD threshold for cell viability. * *p* < 0.05 vs. 0 µM; ** *p* < 0.01 vs. 0 µM; **** *p* < 0.0001 vs. 0 µM.

**Figure 3 ijms-26-05084-f003:**
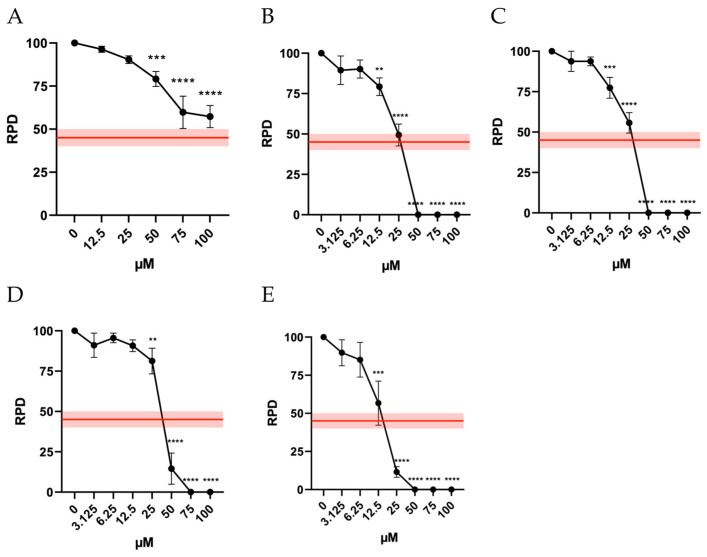
RPD of TK6 cells after 26 h treatment with (**A**) ORF, (**B**) Br-ORF, (**C**) I-ORF, (**D**) F-ORF, and (**E**) Cl-ORF at the concentrations reported compared to the untreated negative control (0 µM). Each value represents the mean ± SEM of at least three independent experiments. Data were analyzed by Repeated Measures ANOVA—(**A**) F (5, 19) = 24.78; *p* < 0.0001; (**B**) F (8, 18) = 52.13; *p* < 0.0001; (**C**) F (8, 18) = 52.52; *p* < 0.0001; (**D**) F (8, 18) = 43.34; *p* < 0.0001; (**E**) F (9, 20) = 12.79; *p* < 0.0001—followed by Dunnett post-test. The red line indicates the OECD threshold for cell replication. ** *p* < 0.01 vs. (0 µM); *** *p* < 0.001 vs. (0 µM); **** *p* < 0.0001 vs. (0 µM).

**Figure 4 ijms-26-05084-f004:**
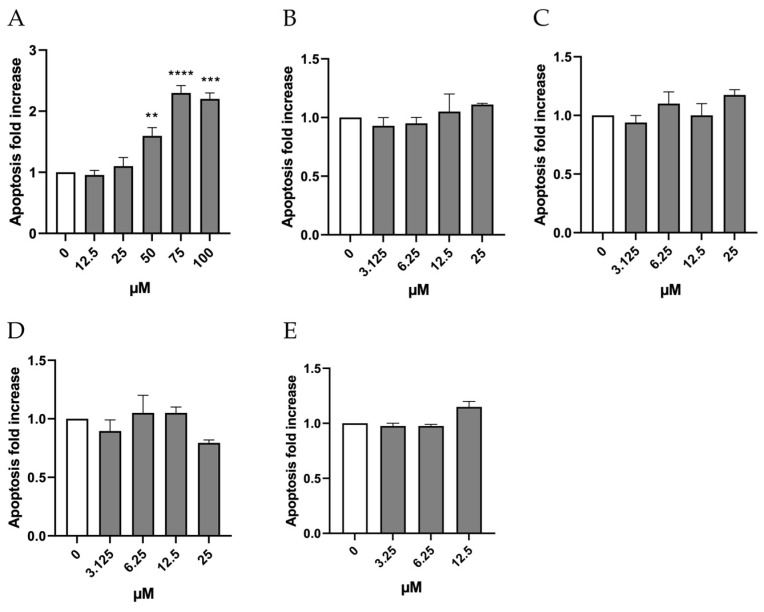
Apoptosis fold increase of TK6 cells after 26 h treatment with (**A**) ORF, (**B**) Br-ORF, (**C**) I-ORF, (**D**) F-ORF and (**E**) Cl-ORF at the concentrations reported compared to the untreated negative control (0 µM). Each value represents the mean ± SEM of at least three independent experiments. Data were analyzed by Repeated Measures ANOVA—(**A**) F (5, 7) = 40.64; *p* < 0.0001; (**B**) F (4, 4) = 0.7246; *p* = 0.6187; (**C**) F (4, 4) = 1.494; *p* = 0.3535; (**D**) F (4, 4) = 2.453; *p* = 0.2030; (**E**) F (3, 3) = 16.04; *p* = 0.0237—followed by Bonferroni or Dunnett post-tests. ** *p* < 0.01 vs. (0 µM); *** *p* < 0.001 vs. (0 µM); **** *p* < 0.0001 vs. (0 µM).

**Figure 5 ijms-26-05084-f005:**
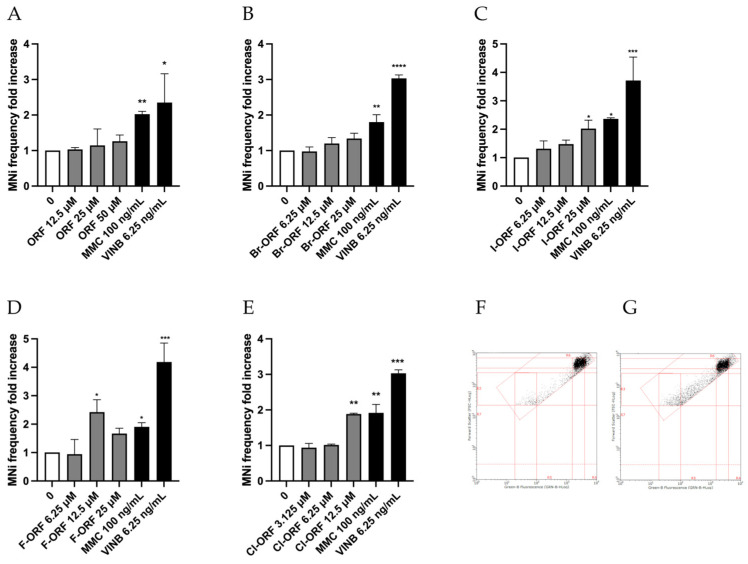
MNi frequency fold increase on TK6 cells after 26 h treatment with (**A**) ORF, (**B**) Br-ORF, (**C**) I-ORF, (**D**) F-ORF and (**E**) Cl-ORF at the concentrations reported compared to the untreated negative control (0 µM) and positive controls (MMC and VINB). Each value represents the mean ± SEM of at least three independent experiments. Data were analyzed by Repeated Measures ANOVA—(**A**) F (5, 11) = 3.157; *p* = 0.0521; (**B**) F (6, 12) = 17.49; *p* < 0.0001; (**C**) F (7, 13) = 11.11; *p* < 0.0001; (**D**) F (6, 13) = 11.01; *p* = 0.0002; (**E**) F (6, 9) = 34.99; *p* < 0.0001—followed by Dunnett post-tests. * *p* < 0.05 vs. (0 µM); ** *p* < 0.01 vs. (0 µM); *** *p* < 0.001 vs. (0 µM); **** *p* < 0.0001 vs. (0 µM). Representative FCM dot plots of nuclei and MNi scored in (**F**) a negative control and (**G**) in 12.5 µM F-ORF-treated cells.

**Figure 6 ijms-26-05084-f006:**
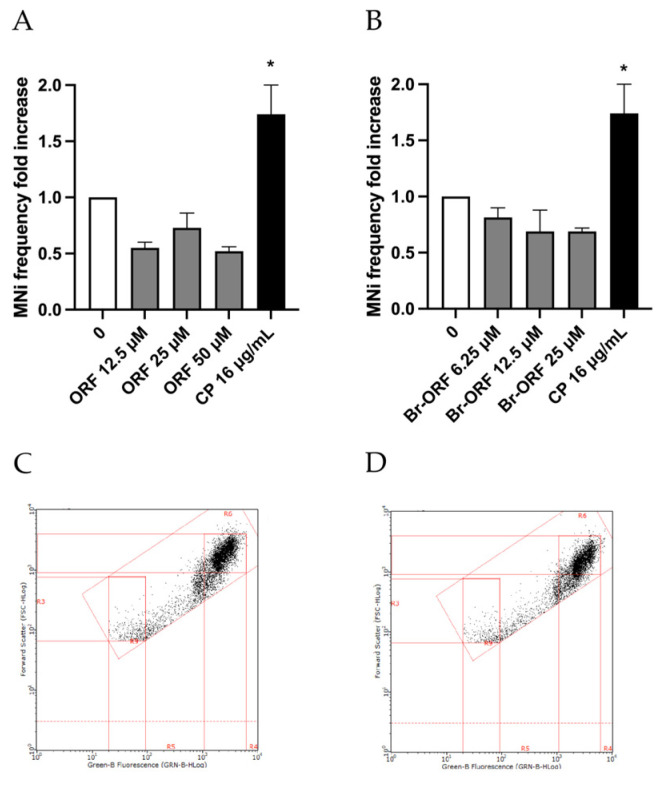
MNi frequency fold increase on TK6 cells after 3 h treatment in the presence of S9 mix with (**A**) ORF or (**B**) Br-ORF at the concentrations reported compared to the untreated negative control (0 µM) and positive control (CP). Each value represents the mean ± SEM of at least three independent experiments. Data were analyzed by Repeated Measures ANOVA—(**A**) F (4, 4) = 11.94; *p* = 0.0170; (**B**) F (4, 4) = 9.590; *p* = 0.0251—followed by Dunnett post-tests. * *p* < 0.05 vs. (0 µM). Representative FCM dot plots of nuclei and MNi scored in (**C**) a negative control and (**D**) in 50 µM ORF-treated cells.

## Data Availability

The data presented in this study are available within the article.

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
