# Peer review of "New Synthetic Opioids: What Do We Know About the Mutagenicity of Brorphine and Its Analogues?"

_ijms, 2025, doi:10.3390/ijms26115084_

Round 1

Reviewer 1 Report

Comments and Suggestions for Authors

This is  a simple study, aiming at evaluating the genotoxicity of a relatively new opioid-like substance, brorphine and its 4 analogs (chemical structures of these compounds could be useful). The whole study consists of the results of 4 tests. The first three tests (viability, proliferation, induction of apoptosis) were performed to find the concentrations of analogs that should be used in the mutagenesis test (MNi frequency).  So the results of this test are actually the only new data obtained in this study. The Authors indicate what further studies should be performed in order to confirm their resuls but they do not perform them. Therefore, it is not surprising that Conclusions contain only one sentence.

In my opinion this manuscript should be presented as a short communication and not a full paper.

Author Response

This is  a simple study, aiming at evaluating the genotoxicity of a relatively new opioid-like substance, brorphine and its 4 analogs (chemical structures of these compounds could be useful). The whole study consists of the results of 4 tests. The first three tests (viability, proliferation, induction of apoptosis) were performed to find the concentrations of analogs that should be used in the mutagenesis test (MNi frequency).  So the results of this test are actually the only new data obtained in this study. The Authors indicate what further studies should be performed in order to confirm their resuls but they do not perform them. Therefore, it is not surprising that Conclusions contain only one sentence.

In my opinion this manuscript should be presented as a short communication and not a full paper

We thank the reviewer for the helpful suggestions to improve our work. The required information and changes are listed point by point and highlighted in red in the manuscript for faster viewing.

In this study, four toxicological endpoints were analysed across four different molecules. For these substances, none of these results have been published to date, making all findings novel and of great relevance for initiating the toxicological profiling of these compounds. They were found to be cytotoxic, both in terms of inducing necrosis and apoptosis, as well as cytostatic. These results are fundamentally important on their own and are also necessary for determining the appropriate doses to be tested in mutagenicity assessments. Nevertheless, the final outcome regarding mutagenic potential remains the most critical, as it highlights the possibility of serious long-term effects.

Moreover, as we stated in the manuscript, to confirm the negative outcome obtained on Orphine and Brorphine further studies should be performed, but for the other three studied molecules supplementary assays are not required: “Future investigations could be aimed at further assessments of the genotoxic potential of Orphine and Brorphine. In fact, for a comprehensive evaluation of the genotoxic hazard and to confirm a negative outcome, an additional assay for genic mutations (e.g., a bacterial reverse mutation test) should be performed, as the MN test only evaluates the ability to cause structural and numerical chromosomal aberrations. On the contrary, in the case of an already positive response for other validated genotoxicity endpoints, e.g., the in vitro mammalian cell MN test, supplementary assays are not required, as for Chlorphine, Iodorphine and Fluorphine”.

So, the toxicological profile of these substances certainly will need further investigation, but that goes beyond the scope of the present study.

Therefore, we do not consider it appropriate to suggest the publication of this study as a short communication, especially considering that our group has already published full papers with similar experimental designs aimed at assessing the mutagenicity of various classes of NPS.

To address the reviewer’s comment, we insert the chemical structure requested (Fig.1) and we implemented the conclusion section.

Reviewer 2 Report

Comments and Suggestions for Authors

This manuscript addresses the urgent topic of genotoxicity associated with novel synthetic opioids (NSOs), focusing on Brorphine and its analogues. The study is methodologically solid, applying OECD-compliant protocols including the in vitro micronucleus test, cytotoxicity evaluation, and metabolic activation (S9). The use of flow cytometry improves objectivity and sensitivity.

However, there are several limitations that need to be addressed to improve the robustness and interpretability of the findings.

Major Concerns:

  1. Single Cell Line Limitation:
    The study was conducted solely on TK6 lymphoblastoid cells. This narrows the biological relevance of the results. Inclusion of at least one additional human-derived cell line (e.g., hepatocytes or epithelial cells) would increase confidence in the generalizability of the findings.
  2. Lack of In Vivo studies:
    While in vitro models are valuable for preliminary screening, the absence of any in vivo studies severely limits translational potential. This should be explicitly acknowledged as a major limitation.
  3. Statistical Analysis Weakness:
    The statistical reporting is insufficient. The authors should report the main effect of drug treatment using classical ANOVA output: F-values with degrees of freedom and associated p-values, prior to any post hoc or pairwise comparisons. This would strengthen the credibility of statistical claims and aid readers in assessing the overall effect sizes.
  4. Lack of Comparative Control (e.g., Fentanyl):
    It is surprising that the study does not include fentanyl or a known opioid analogue as a benchmark. This omission weakens the context of the findings. A comparative control would provide a useful reference point to interpret the genotoxic potential of Brorphine-related compounds.
  5. No Justification of Tested Concentrations vs. Human Exposure:
    The in vitro concentrations used (up to 100 µM) greatly exceed typical plasma levels seen in therapeutic or toxic opioid exposures. The authors must clarify how these concentrations relate to realistic human exposure scenarios to support the biological relevance of their findings.

Author Response

We thank the reviewer for the helpful suggestions to improve our work. The required information and changes are listed point by point and highlighted in red in the manuscript for faster viewing.

  • Single cell line limitation:

The study was conducted solely on TK6 lymphoblastoid cells. This narrows the biological relevance of the results. Inclusion of at least one additional human-derived cell line (e.g., hepatocytes or epithelial cells) would increase confidence in the generalizability of the findings.

OECD guideline 487 specifies that only rodent cell lines such as CHO, V79, CHL/IU, and L5178Y cells or human cell lines such as TK6 can be used to obtain reliable and robust results. Among these, the TK6 cell line was selected as it is the only one of human and non-tumor origin.  We better specified this concept in material and method section. Lines 284-287.

While we acknowledge that incorporating another cell line (e.g., hepatocytes or epithelial cells) could provide broader biological context, we believe that such addition would not significantly alter or strengthen the central conclusions of this study. Moreover, conducting further experiments with additional cell types would entail substantial costs and resource investment, which we do not believe is justified given the clear and consistent results already obtained in the TK6 model.

  • Lack of in vivo studies:

While in vitro models are valuable for preliminary screening, the absence of any in vivo studies severely limits translational potential. This should be explicitly acknowledged as a major limitation.

The execution of in vivo experiments or the concordance with literature data obtained in vivo would certainly strengthen the results of this study, but such data are currently not available. On the other hand, the assessment of a substance’s mutagenicity is one of the few toxicological endpoints for which several in vitro tests have been validated. Furthermore, many regulations agree that in the case of positive results obtained in vitro, these may be considered sufficient, in line with the 3Rs principle and the aim of reducing animal testing as much as possible. In contrast, in vivo experiments are clearly necessary to confirm negative in vitro results, following a precautionary approach to protect human health. To address the reviewer’s comment, a sentence stating that in vivo data would strengthen our results has been included in the result and discussion section. Lines 243-252.

  • Statistical Analysis Weakness:

The statistical reporting is insufficient. The authors should report the main effect of drug treatment using classical ANOVA output: F-values with degrees of freedom and associated p-values, prior to any post hoc or pairwise comparisons. This would strengthen the credibility of statistical claims and aid readers in assessing the overall effect sizes.

As suggested, we insert it.

  • Lack of Comparative Control (e.g., Fentanyl):

It is surprising that the study does not include fentanyl or a known opioid analogue as a benchmark. This omission weakens the context of the findings. A comparative control would provide a useful reference point to interpret the genotoxic potential of Brorphine-related compounds.

As already stated in the abstract, the brorphines analysed in this study are synthetic opioids that are not fentanyl analogues; therefore, a comparison with this compound would not be appropriate. Moreover, fentanyl and three of its analogues were the subject of a previous study published by our group (https://doi.org/10.3390/ijms232214406). Since an appropriate control was not available to address the reviewer’s request, we have included a comparative comment between fentanyl analogues and non-analogues opioids in the result and discussion section. Lines 196-200.

  • No Justification of Tested Concentrations vs. Human Exposure:

The in vitro concentrations used (up to 100 µM) greatly exceed typical plasma levels seen in therapeutic or toxic opioid exposures. The authors must clarify how these concentrations relate to realistic human exposure scenarios to support the biological relevance of their findings.

Unfortunately, information regarding human exposure levels to brorphines is currently not available. However, it is recognized that it is not possible to define a No-Observed-Adverse-Effect Level (NOAEL) for genotoxic substances and zero risk corresponds only to the zero dose, therefore, any dose is potentially toxic. Increasing the dose and the exposure, the likelihood that damage will occur increases. To address the reviewer’s comment, a sentence has been included in the lines 253-261.

Round 2

Reviewer 1 Report

Comments and Suggestions for Authors

The manuscript was improved slightly, though the Conclusion section is weak.

Reviewer 2 Report

Comments and Suggestions for Authors

I accept the revised manuscript and approve it for further processing.